# Heparin-binding motif mutations of human diamine oxidase allow the development of a first-in-class histamine-degrading biopharmaceutical

Elisabeth Gludovacz[1,2], Kornelia Schuetzenberger[3], Marlene Resch[2], Katharina Tillmann[4], Karin Petroczi[2], Markus Schosserer[1], Sigrid Vondra[5], Serhii Vakal[6], Gerald Klanert[1], Jürgen Pollheimer[5], Tiina A Salminen[6], Bernd Jilma[2], Nicole Borth[1], Thomas Boehm[2]*

[1]Department of Biotechnology, University of Natural Resources and Life Sciences, Vienna, Austria; [2]Department of Clinical Pharmacology, Medical University of Vienna, Vienna, Austria; [3]Center for Medical Physics and Biomedical Engineering, Medical University of Vienna, Vienna, Austria; [4]Center for Biomedical Research, Medical University of Vienna, Vienna, Austria; [5]Department of Obstetrics and Gynecology, Medical University of Vienna, Vienna, Austria; [6]Strutural Bioinformatics Laboratory, Biochemistry, Faculty of Science and Engineering, Åbo Akademi University, Turku, Finland

*For correspondence:
thomas.boehm@meduniwien.
ac.at

## Abstract

**Background:** Excessive plasma histamine concentrations cause symptoms in mast cell activation syndrome, mastocytosis, or anaphylaxis. Anti-histamines are often insufficiently efficacious. Human diamine oxidase (hDAO) can rapidly degrade histamine and therefore represents a promising new treatment strategy for conditions with pathological histamine concentrations.

**Methods:** Positively charged amino acids of the heparin-binding motif of hDAO were replaced with polar serine or threonine residues. Binding to heparin and heparan sulfate, cellular internalization and clearance in rodents were examined.

**Results:** Recombinant hDAO is rapidly cleared from the circulation in rats and mice. After mutation of the heparin-binding motif, binding to heparin and heparan sulfate was strongly reduced. The double mutant rhDAO-R568S/R571T showed minimal cellular uptake. The short α-distribution half-life of the wildtype protein was eliminated, and the clearance was significantly reduced in rodents.

**Conclusions:** The successful decrease in plasma clearance of rhDAO by mutations of the heparin-binding motif with unchanged histamine-degrading activity represents the first step towards the development of rhDAO as a first-in-class biopharmaceutical to effectively treat diseases characterized by excessive histamine concentrations in plasma and tissues.

**Funding:** Austrian Science Fund (FWF) Hertha Firnberg program grant T1135 (EG); Sigrid Juselius Foundation, Medicinska Understödsförening Liv och Hälsa rft (TAS and SeV).

## Introduction

The biogenic amine histamine (2-[4-imidazolyl]ethylamine) is stored by basophils and mast cells (MCs) and is rapidly released after stimulation from intracellular storage vesicles. Following activation of the histamine receptors 1 and 2, histamine acts mainly on vascular endothelial, bronchial, and smooth

muscle cells and causes symptoms affecting the cardiovascular system, the skin, and the gastrointestinal and respiratory tracts (*Jutel et al., 2009*; *Valent et al., 2011*). It induces nitric oxide synthesis, which causes vasodilation, vascular hyperpermeability, edema formation, and consequently hypotension (*Ashina et al., 2015*; *Claesson-Welsh, 2015*; *Pober and Sessa, 2015*; *Valent et al., 2019*). The development of hypotension strongly correlates with histamine concentrations (*Kaliner et al., 1982*). Multiple potentially life-threatening conditions such as mast cell activation syndrome (MCAS) or anaphylaxis are accompanied by elevated histamine levels (*Valent et al., 2011*). In severe anaphylaxis, mean histamine concentrations of 140 ng/mL have been measured (*van der Linden et al., 1992*).

The copper-containing amine oxidase human diamine oxidase (hDAO; E.C. 1.4.3.6) is a homodimer encoded by the *AOC1* gene (*Barbry et al., 1990*; *Chassande et al., 1994*), which is highly expressed in the intestine (*Wollin et al., 1998*), the kidneys (*Schwelberger et al., 1998*), and extravillous trophoblasts (*Velicky et al., 2018*). In pregnancy serum, hDAO at a mean concentration of 125 ng/mL rapidly degraded 100 ng/mL histamine with a half-life of 3.4 min (*Boehm et al., 2019*). In the 1930s, porcine diamine oxidase was already used to treat histamine-related pathologies with limited success (*Laymon and Cumming, 1939*; *Miller and Piness, 1940*). After intravenous infusion of recombinant hDAO (rhDAO) in rats and mice, the half-life was less than 5 min (*Gludovacz et al., 2020*). The rapid clearance was not caused by binding of the N-glycans to the asialoglycoprotein- or the mannose-receptor. Diamine oxidase did not only bind to but was also internalized by various epithelial and endothelial cell lines. Internalization was inhibited by high molecular weight heparin (HMWH), and glycosaminoglycan- and heparan sulfate (HS)-deficient CHO cell lines were incapable of rhDAO uptake. Diamine oxidase is produced by intestinal and proximal tubular kidney epithelial cells and binds to the basolateral membranes in the lamina propria after secretion into the interstitial compartment and in vitro also to endothelial cells (*Schwelberger et al., 1999*; *Wollin et al., 1998*). It is released from its binding sites by heparin infusion, resulting in an increase in plasma activity (*Biebl et al., 2002*; *D'Agostino et al., 1989*; *Gäng et al., 1975*; *Hansson and Thysell, 1973*; *Klocker et al., 2004*; *Klocker et al., 2000*). Robinson-White et al. described the binding of rat placental DAO to rat and guinea pig microvascular endothelial cells and its displacement by heparin, indicating that DAO binds to cell surface heparan sulfate proteoglycans (HSPG) (*Robinson-White et al., 1985*). During anaphylaxis, secretion of heparin from MCs likely results in DAO release from HSPG located in the extracellular matrix or on the surface of gastrointestinal epithelial cells (*Boehm et al., 2019*). Cell surface HSPG probably mediate cellular binding and/or uptake of rhDAO.

*Novotny et al., 1994* proposed that the amino acids 568–575 or RFKRKLPK constitute a heparin-binding motif (HBM) (*Novotny et al., 1994*). These amino acid stretches form a ring-like structure including the motifs of both monomers on the surface of DAO (*McGrath et al., 2009*). No negatively charged amino acids are close to this positively charged ring structure (*McGrath et al., 2009*). The putative heparin-binding site of hDAO possesses the size, shape, and electrostatic properties suitable for heparin and HS binding (*Gludovacz et al., 2020*). However, it has never been demonstrated, neither in vitro nor in vivo, that the RFKRKLPK sequence mediates heparin and HS binding.

The short half-life of rhDAO in rats and mice is highly unfavorable for the development of rhDAO as a new treatment option for diseases with rapid and excessive histamine release from MCs or basophils. We herein demonstrate for the first time that rhDAO rapidly degrades pathophysiologically relevant histamine concentrations. Using appropriate mutations in the RFKRKLPK sequence, we prove the heparin binding function of this ring structure motif and its involvement in the ultra-rapid plasma clearance of rhDAO. The HBM mutants show a strongly decreased binding to heparin and HS, minimal cellular internalization, and a strong reduction of the in vivo clearance in rodents with unchanged catalytic activity. This study provides promising data for the development of rhDAO as a first-in-class biopharmaceutical for the treatment of conditions with elevated systemic histamine concentrations.

## Materials and methods

**Key resources table**

| Reagent type (species) or resource | Designation | Source or reference | Identifiers | Additional information |
|---|---|---|---|---|
| Gene (*Homo sapiens*) | AOC1 | GenBank | HGNC: 80 | |

*Continued on next page*

*Continued*

| | | | | |
|---|---|---|---|---|
| Recombinant DNA reagent | pRMCE-hDAO (plasmid) | *Gludovacz et al., 2016* | | hDAO-WT expression vector |
| Recombinant DNA reagent | pRMCE-hDAO-R568S (plasmid) | This paper | | R568S mutant of hDAO |
| Recombinant DNA reagent | pRMCE-hDAO-R571T (plasmid) | This paper | | R571T mutant of hDAO |
| Recombinant DNA reagent | pRMCE-hDAO-K575T (plasmid) | This paper | | K575T mutant of hDAO |
| Recombinant DNA reagent | pRMCE-hDAO-R568S/R571T (plasmid) | This paper | | R568S/R571T mutant of hDAO |
| Recombinant DNA reagent | pRMCE-hDAO-R568S/K575T (plasmid) | This paper | | R568S/K575T mutant of hDAO |
| Recombinant DNA reagent | pRMCE-hDAO-R571T/K575T (plasmid) | This paper | | R571T/K575T mutant of hDAO |
| Recombinant DNA reagent | pRMCE-hDAO- K570G/R571Q/ K572T (plasmid) | This paper | | K570G/R571Q/K572T mutant of hDAO |
| Recombinant DNA reagent | pRMCE-hFcDAO (plasmid) | This paper | | Fusion of human IgG Fc and hDAO |
| Recombinant DNA reagent | pRMCE-hFcDAO-R568S/R571T (plasmid) | This paper | | Fusion of human IgG Fc and R568S/R571T mutant of hDAO |
| Sequence-based reagent | R568S-FP | This paper | PCR primers | ttcaaaaggaagctgccc |
| Sequence-based reagent | R568S-RP | This paper | PCR primers | gctgaaggccgcctggc |
| Sequence-based reagent | R571T-FP | This paper | PCR primers | acgaagctgcccaagtacc |
| Sequence-based reagent | R571T-RP | This paper | PCR primers | tttgaagcggaaggc |
| Sequence-based reagent | K575T-FP | This paper | PCR primers | acgtacctgctctttaccagcc |
| Sequence-based reagent | K575T-RP | This paper | PCR primers | gggcagcttccttttga |
| Sequence-based reagent | R568S/R571T-FP | This paper | PCR primers | aaaacgaagctgcccaagtacctg |
| Sequence-based reagent | R568S/R571T-RP | This paper | PCR primers | gaagctgaaggccgcctgg |
| Sequence-based reagent | R571T/K575T-FP | This paper | PCR primers | gcccacgtacctgctctttaccagccc |
| Sequence-based reagent | R571T/K575T-RP | This paper | PCR primers | agcttcgttttgaagcggaaggc |
| Sequence-based reagent | K570G/R571Q/K572T-FP | This paper | PCR primers | cagacgctgcccaagtacctgct |
| Sequence-based reagent | K570G/R571Q/K572T-RP | This paper | PCR primers | tccgaagcggaaggccg |
| Peptide, recombinant protein | rhDAO-WT | *Gludovacz et al., 2016* | | Recombinant human diamine oxidase, wildtype |
| Peptide, recombinant protein | rhDAO-R568S | This paper | | R568S mutant of rhDAO-WT |
| Peptide, recombinant protein | rhDAO-K575T | This paper | | K575T mutant of rhDAO-WT |

*Continued on next page*

*Continued*

| | | | |
|---|---|---|---|
| Peptide, recombinant protein | rhDAO-R568S/R571T | This paper | R568S/R571T mutant of rhDAO-WT |
| Peptide, recombinant protein | rhDAO-R568S/K575T | This paper | R568S/K575T mutant of rhDAO-WT |
| Peptide, recombinant protein | rhDAO- K570G/R571Q/K572T | This paper | K570G/R571Q/K572T mutant of rhDAO-WT |
| Peptide, recombinant protein | rhFcDAO | This paper | Human IgG Fc-rhDAO fusion protein |
| Peptide, recombinant protein | rhFcDAO-R568S/R571T | This paper | Human IgG Fc-rhDAO-R568S/R571T fusion protein |
| Cell line (*Cricetulus griseus*) | CHO-K1_rhDAO-WT | *Gludovacz et al., 2016* | CHO-K1 stably expressing rhDAO-WT |
| Cell line (*C. griseus*) | CHO-K1_rhDAO-R568S | This paper | CHO-K1 stably expressing rhDAO-R568S |
| Cell line (*C. griseus*) | CHO-K1_rhDAO-K575T | This paper | CHO-K1 stably expressing rhDAO-K575T |
| Cell line (*C. griseus*) | CHO-K1_rhDAO-R568S/R571T | This paper | CHO-K1 stably expressing rhDAO-R568S/R571T |
| Cell line (*C. griseus*) | CHO-K1_rhDAO-R567S/K575T | This paper | CHO-K1 stably expressing rhDAO-R567S/K575T |
| Cell line (*C. griseus*) | CHO-K1_rhDAO- K570G/ R571Q/K572T | This paper | CHO-K1 stably expressing rhDAO-K570G/R571Q/K572T |
| Cell line (*C. griseus*) | CHO-K1_rhFcDAO | This paper | CHO-K1 stably expressing rhFcDAO |
| Cell line (*C. griseus*) | CHO-K1_ rhFcDAO-R568S/R571T | This paper | CHO-K1 stably expressing rhFcDAO-R568S/R571T |
| Cell line (*C. griseus*) | ExpiCHO-S | Thermo Fisher Scientific | Cat#: A29133 RRID:CVCL_5J31 |
| Cell line (*C. griseus*) | CHO-K1 | ATCC | Cat#: CCL-61 RRID:CVCL_0214 |
| Cell line (*H. sapiens*) | SK-Hep1 | Sigma-Aldrich | Cat#: 91091816 RRID:CVCL_0525 |
| Cell line (*H. sapiens*) | HUVEC67 | Evercyte | Cat#: CPT-006-0067 |
| Cell line (*H. sapiens*) | HUVEC/TERT2 | Evercyte | Cat#: CHT-006-0008 RRID:CVCL_9Q53 |
| Cell line (*H. sapiens*) | HDMVEC/TERT164-B | Evercyte | Cat#: CHT-013-0164-B |
| Cell line (*H. sapiens*) | HDF76 | Evercyte | Cat#: CPT-008-0076 |
| Cell line (*H. sapiens*) | PODO/TERT256 | Evercyte | Cat#: CHT-033-0256 RRID:CVCL_JL76 |
| Cell line (*H. sapiens*) | LHCN-M2 | Evercyte | Cat#: CkHT-040-231-2 RRID:CVCL_8890 |
| Cell line (*H. sapiens*) | HepG2 | ATCC | Cat#: HB-8065 RRID:CVCL_0027 |
| Cell line (*H. sapiens*) | HeLa | Ellmeier Lab – Medical University of Vienna | |

*Continued on next page*

*Continued*

| | | | | |
|---|---|---|---|---|
| Cell line (*H. sapiens*) | EVT | *Velicky et al., 2018* | | |
| Antibody | Anti-ABP1 (rabbit polyclonal) | Sigma-Aldrich | Cat#: SAB1410491-100UG | IF (1:500) |
| Antibody | Alexa Fluor 488 anti-rabbit (H + L) (donkey polyclonal) | Jackson Research | Cat#: 711-545-152 RRID:AB_2313584 | IF (1:500) |
| Antibody | IgG serum fraction (rabbit polyclonal) | *Boehm et al., 2017* | | WB (1:1000) |
| Antibody | β-actin mAB (AC-15) (mouse monoclonal) | Invitrogen | Cat#: AM4302 RRID:AB_2536382 | WB (1:5000) |
| Antibody | IRDye 800CW anti-rabbit IgG (H + L) (goat polyclonal) | Li-Cor | Cat#: 926-32211 RRID:AB_621843 | WB (1:5000) |
| Antibody | IRDye 680RD anti-mouse IgG (H + L) (goat polyclonal) | Li-Cor | Cat#: 925-68070 RRID:AB_2651128 | WB (1:5000) |
| Chemical compound, drug | Histamine | Sigma-Aldrich | Cat#: 53300 | |
| Chemical compound, drug | Heparin | Gilvasan | 1000 IU/mL | |
| Chemical compound, drug | Heparin (sodium salt from intestinal mucosa) | Sigma-Aldrich | Cat#: H3149 | |
| Chemical compound, drug | Diminazene aceturate | Sigma-Aldrich | Cat#: D7770 | |
| Commercial assay or kit | Cisbio HTRF histamine 500 test kit | Biomedica | Cat#: 62HTMDPET | |
| Commercial assay or kit | Immunotech histamine ELISA | Beckman Coulter | Cat#: IM2562 | |
| Commercial assay or kit | ExpiCHO Expression System Kit | Thermo Fisher Scientific | Cat#: A29133 | |
| Software, algorithm | R (version 3.6.3) | *R Development Core Team, 2020* | | |
| Software, algorithm | Kaluza Flow Cytometry Analysis Software (version 2.1) | Beckman Coulter | | |
| Software, algorithm | ImageJ/Fiji | *Schneider et al., 2012* | | |
| Software, algorithm | Pymol (version 2.0) | The PyMOL Molecular Graphics System, version 2.0 Schrödinger, LLC | | |
| Software, algorithm | Chimera | UCSF | | AmberTools GUI incorporated |
| Software, algorithm | BIOVIA Discovery Studio 2019 | Dassault Systemes | | Modeller 9.2 GUI incorporated |
| Software, algorithm | FoldX 4.0 | *Schymkowitz et al., 2005* | | |
| Other | DAPI stain | Invitrogen | Cat#: D1306 | (80 ng/mL) |

## Examination of histamine degradation

Degradation of 100 ng/mL histamine (Sigma-Aldrich, St. Louis, MO, 53300) by 6 nM (1 µg/mL) rhDAO variants was measured in duplicate in phosphate buffered saline (PBS) with 1% human serum albumin (HSA; Albunorm; Octapharma, Vienna, Austria) or in EDTA plasma from a healthy volunteer, with or without 600 nM HMWH added (heparin, 1000 IU/mL equivalent to 2 mg/mL, Gilvasan, Vienna, Austria), using a Cisbio HTRF histamine 500 test kit (Biomedica, Vienna, Austria).

Plasma of seven mastocytosis patients and three healthy volunteers was spiked with 6 nM rhDAO-WT, 545 nM histamine, and 20 µM of the potent DAO-inhibitor diminazene aceturate (DIMAZ; Sigma-Aldrich, D7770) and histamine concentrations were determined in duplicate with a histamine ELISA (Immunotech IM2562, Beckman Coulter, Brea, CA) as described previously (*Boehm et al., 2019*).

## Site-directed mutagenesis to create HBM mutants

To generate rhDAO single and double HBM mutants, the charged amino acids arginine and lysine were replaced with polar threonine and serine by site-directed mutagenesis. Threonine and serine have been used to replace arginine residues in the HBM of fibronectin, resulting in significantly reduced binding affinities, while at the same time retaining the three-dimensional structure (*Busby et al., 1995*; *Kapila et al., 2001*). The triple mutant rhDAO-K570G/R571Q/K572T representing the HBM of guinea pig, dog, rat, mouse, and Chinese hamster DAO was also tested. All mutants were generated from the rhDAO expression plasmid described by *Gludovacz et al., 2016*. The HBM mutants and the respective 5'-phosphorylated primers are summarized in the Key resources table. The cloning procedure is described by *Gludovacz et al., 2018*. Human IgG Fc was fused to the N-termini of rhDAO-WT (rhFcDAO) and rhDAO-R568S/R571T (rhFcDAO-R568S/R571T) using an IgG1 hinge region as a linker. The Fc sequence was synthesized by Eurofins MWG Operon (Ebersberg, Germany), and plasmids were constructed by using the methods described by *Gludovacz et al., 2016*.

## Expression and purification of the HBM mutants

Stable CHO-K1 cell lines expressing the different HBM mutants were generated and cultivated as described for the wildtype (*Gludovacz et al., 2016*). rhDAO-R571T and -R571T/K575T could not be successfully expressed. For batch cultivation, cells were seeded at a viable cell density of 0.2 × 10$^6$ cells/mL and incubated for 8 days. 10 µM CuSO$_4$ was added on days 0 and 4. One batch of rhDAO-WT (batch 2) and rhDAO-R568S/R571T (batch 3) was produced with the ExpiCHO expression system (Thermo Fisher Scientific, Waltham, MA) using the standard protocol over 10 days (see 'ExpiCHO Expression System User Guide,' MAN0014337). 10 µM CuSO$_4$ was added on days 0, 2, 4, 6, and 8.

Culture supernatants were ultra- and diafiltrated using the Labscale TFF System in combination with one to three Pellicon XL 50 Ultrafiltration Cassette-Biomax Polyethersulfone with a 100 kDa molecular mass cutoff (both Merck Millipore, Burlington, MA). The supernatants were concentrated 10- to 50-fold, and the culture media were replaced by 10 mM potassium phosphate buffer, pH 7.4 (Merck Millipore). The samples were loaded onto three 5 mL HiTrap Heparin HP columns connected in series at a flow rate of 2 mL/min using an Äkta Purifier or Start HPLC device (all GE Healthcare, Chicago, IL). Stepwise elution of rhDAO-WT, rhFcDAO, and rhDAO-K575T was performed with 0.25, 0.5, and 1 M KCl. The other HBM mutants were eluted with 0.125, 0.25, and 1 M KCl (Sigma-Aldrich) in 10 mM potassium phosphate buffer, pH 7.4. The eluates containing rhDAO variants were concentrated and desalted using the Labscale TFF System combined with one Pellicon XL 50 Ultrafiltration Cassette and 25 mM Tris-HCl buffer, pH 8.5 (AppliChem, Darmstadt, Germany). This material was loaded onto a 5 mL HiTrap CaptoQ column (GE Healthcare) at a flow rate of 3 mL/min. Stepwise elution was conducted with 0.315 and 1.5 M KCl in 25 mM Tris-HCl buffer, pH 8.5. Buffer exchange of the 0.315 M KCl eluate against 50 mM HEPES with 150 mM KCl, pH 7.5 was performed with the Labscale TFF System and one Pellicon XL 50 Ultrafiltration Cassette. Purified rhDAO was quantified using NanoDrop 1000 spectrophotometer (Thermo Fisher Scientific).

## Determination of heparin binding using heparin-sepharose

The storage buffer of purified rhDAO-WT and the HBM mutants was replaced with either 10 mM potassium phosphate, pH 7.4, or 50 mM HEPES, pH 7.4, using Amicon Ultra centrifugal filter units (MWCO 100 kDa, Merck Millipore). They were loaded onto a HiTrap Heparin HP column (1 mL, GE Healthcare) at a flow rate of 0.2 mL/min using an Äkta purifier HPLC device. Elution was performed using a linear gradient of increasing amounts of 1 M KCl in 10 mM potassium phosphate, pH 7.4, or 1 M NaCl in 50 mM HEPES, pH 7.4.

## Analysis of rhDAO-R568S/R571T binding to HMWH and HS

Binding of rhDAO-R568S/R571T to HMWH and HS compared to rhDAO-WT was analyzed with ITC using automated MicroCal PEAQ-ITC (Malvern Panalytical, Malvern, UK) and with BLI using Octet RED96e (FortéBio, Fremont, CA) as described previously (*Gludovacz et al., 2020*). Sample concentrations and injection modes of all ITC experiments are listed in *Figure 2—source data 1*. For BLI comparison of binding of the wildtype protein and the R568S/R571T mutant to HS and HMWH, protein concentrations of 471 nM (HS) and 88 nM (HMWH) were used.

## In vitro cell-based uptake assay

### Cell lines and strains

The following cell lines and strains were used for the experiments in this study: HepG2, CHO-K1, human dermal fibroblasts (HDF), HeLa, SK-Hep1, primary human extravillous trophoblast cells (EVT), primary human umbilical vein endothelial cells (HUVEC67), immortalized HUVECs (HUVEC/TERT2), immortalized human dermal microvascular endothelial cells (HDMVEC/TERT164-B), immortalized podocytes (PODO/TERT256), and immortalized myoblasts (LHCN-M2). All cell lines were incubated in a humidified atmosphere at 37 °C with 5% $CO_2$ and passaged twice a week.

Suppliers, catalog numbers, and culture media compositions were recently published (*Gludovacz et al., 2020*). The identity of all cell lines has been authenticated. Absence of mycoplasma has been confirmed on a regular basis.

### Flow cytometry

The cells were seeded into 96-well plates. At confluency, the supernatant was replaced with 50 µL fresh culture medium supplemented with 30 nM Alexa488-labeled rhDAO-WT or rhDAO-R568S/R571T and incubated for 60 min at 37 °C. These variants were fluorescently labeled by periodation of vicinal diol groups and subsequent conjugation with an Alexa Fluor 488-hydrazide (*Gludovacz et al., 2020*). No DAO added served as a negative control. The cells were washed three times for 5 min with PBS containing 500 µg/mL heparin (sodium salt from intestinal mucosa, Sigma-Aldrich) at room temperature and detached with 50 µL 0.1% trypsin, 0.02% EDTA (Sigma-Aldrich). The reaction was stopped with 50 µL 1× trypsin inhibitor (Sigma-Aldrich, T6414) and addition of 100 µL PBS. 500 cells per cell sample were analyzed on a CytoFLEX S flow cytometer.

Raw data of fluorescence measurements were analyzed with the help of R version 3.6.3 (*R Development Core Team, 2020*) and RStudio 1.1.463 including the following package: userfriendlyscience version 0.7.2. Data of each tested cell line was loaded into R, log-transformed, and normalized between biological replicates by dividing the data by the median of the respective negative controls. The standard deviations of each cell line were adjusted between biological replicates by dividing the standard deviation of the negative controls of the first biological replicate by the standard deviation of the negative controls of the second biological replicate. This factor was then used to adjust the standard deviation of all samples. The medians of all samples were used for statistical testing. First, a Shapiro–Wilk test was used to test whether the medians were normally distributed. As this was the case for all samples and cell lines, a Bartlett test was used to see whether all samples of a cell line have equal variances. If this was the case, a one-way ANOVA with a Tukey's HSD test as a posthoc test was used to identify significant differences. If equality of variances could not be assumed, an ANOVA with Welch's correction and a Games–Howell posthoc test were applied. If a p-value of <0.05 was calculated, significant differences were assumed.

Background corrected means of the median values ± SEM: median fluorescence intensity values of all samples were calculated with Kaluza Flow Cytometry Analysis Software (Beckman Coulter, version 2.1). After correcting for background fluorescence, the means of the medians ± SEM are presented.

### Fluorescence microscopy

The cells were seeded into 8-well µ-slides (ibidi, Martinsried, Germany, 80826) 3–4 days prior to the experiments. They were washed once with PBS and then covered with 130 µL of the respective culture medium containing 60–120 nM unlabeled or Alexa488-labeled rhDAO. Incubation was performed at 37 °C for 60 min. The cells were washed three times for 5 min with PBS containing 500 µg/mL heparin at room temperature. The cell fixation, permeabilization staining, and imaging procedures were recently published (*Gludovacz et al., 2020*).

### Western blot

Internalization of rhDAO wildtype and the HBM mutants R568S, R568S/R571T, and R568S/K575T into SK-Hep1 and HUVEC/TERT2 cells was tested as described (*Gludovacz et al., 2020*).

## Determination of in vivo clearance and DAO antigen and activity

rhDAO-WT and the different mutants were administered to rats and mice as described (*Gludovacz et al., 2020*). DAO antigen concentrations and enzymatic activity were measured in duplicate as published (*Boehm et al., 2020*; *Boehm et al., 2017*). The activity curve in the linear range was used to calculate the slope corresponding to the oxidation rate.

## Ethics statement

The experimental protocols for the treatment of rats and mice were approved by the local Animal Welfare Committee and the Federal Ministry of Science, Research and Economy (GZ 66.009/0152--WF/V/3b/2014) and conducted in full accordance with the ARRIVE guidelines (*Kilkenny et al., 2010*).

## Prediction of mutation effects on protein stability and heparin-binding affinity in silico

The complex of wildtype hDAO and heparin hexasaccharide was constructed as described previously (*Gludovacz et al., 2020*). Briefly, the missing side-chain atoms in the DAO dimer were first built with Pymol (The PyMOL Molecular Graphics System, version 2.0, *Schrödinger, LLC, 2021*). Using ClusPro, heparin tetrasaccharide probes were blindly docked into DAO dimer to reveal the heparin-binding site, which was later used for a restrained docking. Heparin hexasaccharide was then manually built by joining two probes in Pymol and subjected to restrained energy minimization using AmberTools GUI incorporated in Chimera (UCSF).

Three single (R568S, R571T, and K575T), three double (R568S/R571T, R568S/K575T, and R571T/K575T), and one triple (K570G/R571Q/K572T) mutants were built with Modeller 9.2 GUI in BIOVIA Discovery Studio 2019 (Dassault Systemes). Twenty models for each mutant were built without global optimization to avoid driving the conformation too far from the initial one. The model with the lowest probability density function (PDF) total energy and discrete optimized protein energy (DOPE) score was selected as the representative structure. The 'Side-Chain Refinement' protocol was then used to

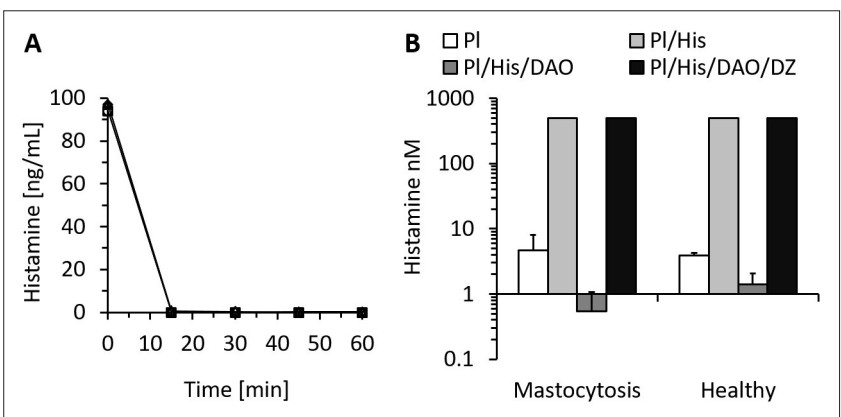

**Figure 1.** Recombinant human diamine oxidase (rhDAO) efficiently degrades histamine in phosphate buffered saline (PBS) and human plasma samples from mastocytosis patients and healthy volunteers. (**A**) 6 nM purified rhDAO-WT with and without 100× high molecular weight heparin (HMWH; 600 nM) were incubated at 37 °C for 60 min with 100 ng/mL histamine in 1% human serum albumin-phosphate buffered saline (HSA-PBS) and EDTA plasma of a healthy volunteer. Histamine concentrations were determined in duplicate. Plasma: △; plasma with 100× HMWH: ◇; HSA-PBS: O; HSA-PBS with 100× HMWH: □. (**B**) Plasma of mastocytosis patients (n = 7) and healthy volunteers (n = 3) was spiked with 6 nM rhDAO-WT, 545 nM histamine, and 20 µM diminazene aceturate (DZ, potent DAO inhibitor) and incubated for 60 min at 37 °C. Mean histamine concentrations with standard deviation (SD) as applicable are shown; column Pl (plasma): no histamine and no DAO spiking but with endogenous histamine; column Pl/His/DAO: spiking of exogenous histamine and exogenous DAO; column Pl/His: spiking of exogenous histamine; column Pl/His/DAO/DZ: spiking of exogenous histamine, DAO, and DIMAZ; t-test p-value comparing Pl with Pl/His/DAO data was 0.015 for mastocytosis patients and 0.011 for healthy volunteers; Histamine concentrations in Pl/His and Pl/His/DAO/DZ samples were >500 nM and the SD could not be calculated. Pl: plasma; His: histamine.

optimize the mutated side chains without affecting the overall fold. For each mutant, the electrostatic surface was calculated with the APBS plugin (*Baker et al., 2001*) and visualized in Pymol.

To predict the effects of the mutations on DAO stability, the FoldX 4.0 (Centre for Genomic Regulation, Spain; *Schymkowitz et al., 2005*) command-line interface and 'PositionScan' command were used. Free energy contributions from individual mutations were summed to get total free energy changes.

Changes in free energy of binding to heparin hexasaccharide were predicted with the 'Calculate mutation energy (Binding)' protocol in Discovery Studio 2019 (*Spassov and Yan, 2013*) using the following parameters: pH-dependent electrostatics = true, pH = 7.4, preliminary minimization = false, forcefield = CHARMm Polar H. All other options were used with default values.

## Results

### rhDAO rapidly degrades endogenous and exogenous histamine

The lowest published $K_m$ of rhDAO for histamine is 2.8 µM or 311 ng/mL (*Elmore et al., 2002*). Histamine concentrations inducing clinically relevant hypotension start at approximately 5 ng/mL, which is 62-fold below the $K_m$. Is rhDAO able to rapidly degrade histamine using a pathophysiologically relevant histamine concentration range of 5–100 ng/mL? 6 nM (1 µg/mL) rhDAO-WT with and without a

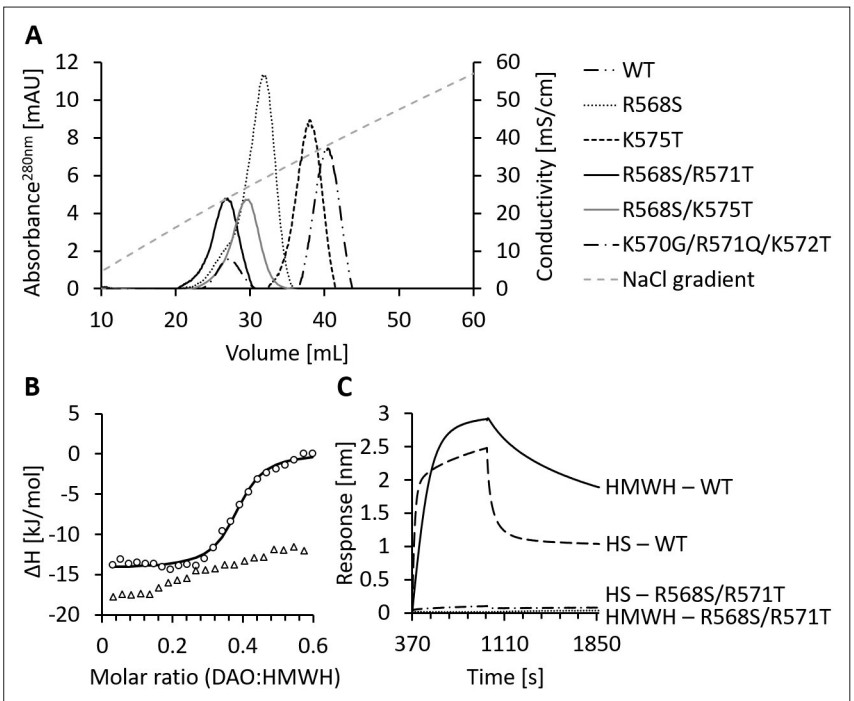

**Figure 2.** Mutations of the heparin-binding motif of recombinant human diamine oxidase (rhDAO) significantly reduce binding to heparin and heparan sulfate (HS). (**A**) Purified rhDAO-WT and various heparin-binding motif (HBM) mutants were loaded onto a heparin-sepharose column and eluted with a linear gradient of increasing NaCl concentration. (**B**) Isothermal titration calorimetry was performed by titrating 25 × 1.5 µL high molecular weight heparin (HMWH) at a concentration of 120 µM into 36.7 µM rhDAO-WT (O) or rhDAO-R568S/R571T (△). The graph represents single experiments for each protein-ligand pair. A summary of all experiments is shown in *Figure 2—source data 1*. (**C**) Biolayer interferometry. Streptavidin sensors loaded with biotinylated HMWH or HS were incubated for 10 min with 88 nM or 471 nM rhDAO, respectively. Dissociation was measured for 15 min. The graphs represent one of three individual measurements and show the association and dissociation curves after subtraction of the negative control (no DAO added). The data of rhDAO-WT have already been published (*Gludovacz et al., 2020*) but are added for better presentation.

The online version of this article includes the following source data for figure 2:

**Source data 1.** All raw plots and integrated heat plots of the isothermal titration calorimetry (ITC) analyses of the heparin-binding motif (HBM) mutant are presented in *Figure 2B*.

**Table 1.** Mutations in the heparin-binding motif of recombinant human diamine oxidase (rhDAO) reduce binding to heparin-sepharose.

| rhDAO | NaCl | | KCl | | Mean (SD) |
|---|---|---|---|---|---|
| | Concentration (mM) | Reduction (mM) | Concentration (mM) | Reduction (mM) | Reduction (mM) |
| WT | 372 | 0 | 310 | 0 | 0 (na) |
| R568S | 186 | 186 | 172 | 138 | 162 (24) |
| K575T | 317 | 55 | 233 | 77 | 66 (11) |
| R568S/R571T | 175 | 197 | 138 | 172 | 185 (13) |
| R568S/K575T | 197 | 175 | 164 | 146 | 161 (15) |
| K570G/R571Q/K572T* | 219 | 153 | - | - | - |

Purified rhDAO-WT and various HBM mutants were loaded onto a heparin-sepharose column and eluted with linear gradients of 0–1 M NaCl or KCl. The salt concentrations necessary for rhDAO elution and the reduction thereof compared to the wildtype protein are shown. The physiological salt concentration in blood plasma and interstitial fluid is approximately 145 mM.

*Corresponds to the HBM in guinea pig, dog, rat, mouse, and Chinese hamster.

HBM = heparin-binding motif. SD = standard deviation. na = not applicable.

100-fold molar excess of HMWH completely degraded 100 ng/mL histamine in less than 15 min using 1% HSA-PBS or EDTA plasma (*Figure 1A*). Heparin did not influence DAO activity. Plasma of seven mastocytosis patients and three healthy volunteers with endogenous histamine levels of approximately 5 nM were also spiked with 545 nM exogenous histamine. rhDAO-WT degraded exogenous and endogenous histamine to harmless concentrations below 1 ng/mL. Histamine deamination was completely blocked by addition of the potent DAO inhibitor DIMAZ (*Figure 1B*).

## The proposed HBM is essential for binding to heparin and heparan sulfate

We have recently shown in rodents that rhDAO-WT is rapidly cleared from the circulation, precluding its use as a histamine-degrading biopharmaceutical (*Gludovacz et al., 2020*). Mutations in the HBM of superoxide dismutase increased the plasma concentration 10-fold (*Sandström et al., 1994*). Based on the superoxide dismutase data and the complete blockade of rhDAO uptake into various cell lines using HMWH (*Gludovacz et al., 2020*), we hypothesized that the proposed HBM of DAO is involved in the rapid clearance of rhDAO in vivo. The tested HBM mutants eluted from the heparin-sepharose at lower salt concentrations than the wildtype protein (*Figure 2* and *Table 1*). rhDAO-R568S/R571T demonstrated the weakest binding to heparin-sepharose and was therefore further analyzed using isothermal titration calorimetry (ITC) and biolayer interferometry (BLI). rhDAO-WT bound to HMWH with mean (SD) $K_D$ values of 634 (26) nM in ITC and 1.6 and 69 nM in BLI analyses with likely two DAO molecules binding to one HMWH molecule (*Gludovacz et al., 2020*). The mean $K_D$ values for HS were 4 and 112 nM using BLI. The best curve fit was again generated with a ratio of two molecules DAO binding to one molecule HS (*Gludovacz et al., 2020*). rhDAO-R568S/R571T, the rhDAO mutant with the lowest heparin affinity, did not show any binding to HMWH and only minimal binding to HS (*Figure 2B and C*). All raw plots and integrated heat plots of the ITC analyses of the HBM mutant are presented in *Figure 2—source data 1*.

## Mutations of the HBM significantly reduce cellular internalization of rhDAO

rhDAO-WT is rapidly internalized into various cell types (*Gludovacz et al., 2020*). To test whether mutations of the HBM not only reduce binding of rhDAO to HMWH and HS, but also inhibit cellular uptake, the single mutant rhDAO-R568S and the two double mutants, rhDAO-R568S/R571T and rhDAO-R568S/K575T, were selected for cellular internalization experiments. SK-Hep1 cells showed a slightly reduced uptake of the single mutant rhDAO-R568S, but a strong decrease for both double mutants (*Figure 3A*). The other cell lines were only incubated with rhDAO-WT and the double mutant

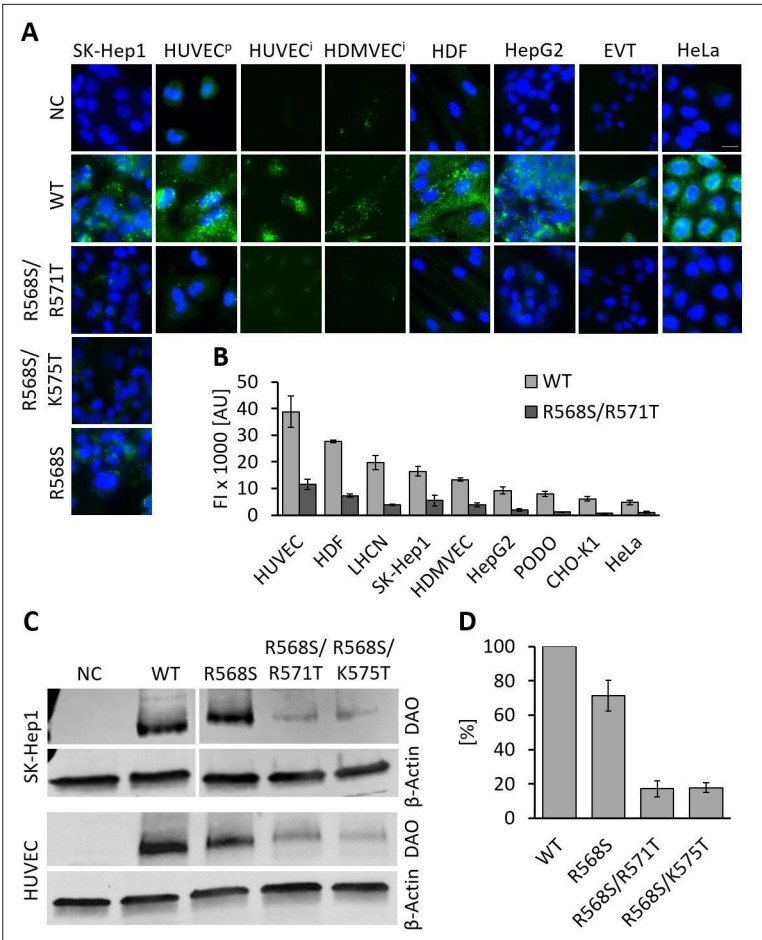

**Figure 3.** Recombinant human diamine oxidase (rhDAO) heparin-binding motif (HBM) double mutants show significantly reduced uptake into various cell lines. (**A**) SK-Hep1, primary human umbilical vein endothelial cells ( = HUVEC[p]), human dermal fibroblasts (HDF), HepG2, and HeLa cells were incubated with 120 nM unlabeled purified rhDAO-WT and various HBM mutants and detected with rabbit anti-ABP1 antibodies and Alexa Fluor 488 donkey anti-rabbit antibodies after fixation and permeabilization. Immortalized HUVEC/TERT2 ( = HUVEC[i]) and human dermal microvascular endothelial cells (HDMVEC/TERT164-B = HDMVEC[i]) were incubated with 120 nM and extravillous trophoblasts (EVT) with 60 nM Alexa488-labeled rhDAO-WT and rhDAO-R568S/R571T; Scale bar = 20 µm. (**B**) Cells were incubated with 30 nM Alexa488-labeled rhDAO-WT and rhDAO-R568S/R571T mutant (no DAO added = negative control), washed, and analyzed flow cytometrically (500 cells per sample). Background corrected mean median values ± standard error of the mean (SEM) are shown (n = 4 biological replicates, two individual experiments in duplicate). (**C**) $10^6$ SK-Hep1 and $5 \times 10^5$ HUVEC/TERT2 cells were incubated with 120 nM and 60 nM rhDAO-WT and three HBM mutants. The cell lysates were analyzed for DAO uptake using western blotting. β-Actin was used as an internal standard. (**D**) After background subtraction, rhDAO band intensities were corrected against β-Actin and normalized against the rhDAO-WT band. The mean ± SEM of two individual experiments with SK-Hep1 cells and one experiment with HUVEC/TERT2 cells are shown. (**A–D**) Incubations were performed for 60 min at 37 °C. FI: fluorescence intensity; AU: arbitrary units.

The online version of this article includes the following source data for figure 3:

**Source data 1.** Statistical evaluation of flow cytometry data is summarized in *Figure 3B*.

**Source data 2.** Raw unedited western blots are presented in *Figure 3C*.

rhDAO-R568S/R571T. The wildtype protein showed intracellular granular staining. The fluorescence signal using the mutant was comparable to the negative controls with no DAO added (*Figure 3A*).

In flow cytometric analyses, the reduction in fluorescence of the double mutant rhDAO-R568S/R571T ranged from 67% in SK-Hep1 to 85% in PODO/TERT256 cells (*Figure 3B*). The decrease in fluorescence intensity was statistically significant for all cell lines (p-value<0.05; *Figure 3—source data 1*).

**Table 2.** Recombinant human diamine oxidase (rhDAO) heparin-binding motif mutants eliminate the fast α-distribution half-life and strongly reduce clearance in rats and mice.

| rhDAO | Rats IV | | | |
| | AUC (µg/mL/min)* | Fold increase | $t_{1/2}$ α (min) | $t_{1/2}$ β (min) |
|---|---|---|---|---|
| WT | 116 | 1.0 | 3.7 | 250 |
| R568S | 1531 | 13.2 | na | 294 |
| R568S/R571T | 3788 | 32.6 | na | 353 |
| R568S/K575T | 2113 | 18.2 | 42 | 361 |
| Fc-WT | 69 | 1.0 | 1.7 | 120 |
| Fc-R568S/R571T | 2723 | 39.5 | na | 268 |

| | Mice IV | | | Mice IP | | |
| | AUC (µg/mL/min)† | Fold increase | $t_{1/2}$ (min)‡ | AUC (µg/mL/min)§ | Fold increase | $t_{1/2}$ (min)‡ |
|---|---|---|---|---|---|---|
| WT | 76 | 1.0 | 76.1 | 41.2 | 1.0 | 200 |
| R568S/R571T | 1468 | 19.4 | 192.1 | 666.3 | 16.2 | 394 |

*Calculated from 5 to 1440 min.
†Calculated from 10 to 1680 min.
‡Calculated from 60 to 1680 min.
§Calculated from 0 to 1440 min.
AUC = area under the curve. IV = intravenous. IP = intraperitoneal. na = not applicable.

Western blot analyses of SK-Hep1 and HUVEC/TERT2 cells incubated with rhDAO-WT and three HBM mutants support the results obtained using fluorescence microscopy and flow cytometry. The mean (SEM) uptake of the single mutant was reduced by only 29–71% (9%), while the band intensities of rhDAO-R568S/R571T and rhDAO-R568S/K575T decreased more than fivefold to 17% (5%) and 18% (3%), respectively (*Figure 3C and D*).

## Clearance of HBM mutants is strongly decreased compared to wildtype DAO in rats and mice

After demonstrating reduced in vitro cellular uptake using the HBM mutants, pharmacokinetic parameters were determined in rats and mice. The data are summarized in *Table 2*. The double mutant rhDAO-R568S/R571T generated the highest area under the curve (AUC), followed by rhDAO-R568S and rhDAO-R568S/K575T (*Figure 4A*). We also tested an rhFc-DAO wildtype fusion protein and the corresponding rhFc-DAO-R568S/R571T mutant confirming the improved pharmacokinetic parameters (*Figure 4B*).

The most promising variant rhDAO-R568S/R571T was also tested in mice, and the data support the rat data with more than 15-fold increases in the AUC after intravenous or intraperitoneal administration (*Figure 4C and D* and *Table 2*).

The rhDAO-R568S/R571T mutant with and without a 100-fold molar excess of HMWH degraded histamine as efficiently as wildtype DAO using 1 µg/mL or 6 nM enzyme concentration (*Figure 5A* compared to WT in *Figure 1A*). No significant differences in enzymatic activity could be detected comparing rhDAO-WT and -R568S/R571T mutant at concentrations of 0.7, 2, and 6 nM using a different assay format (*Figure 5—figure supplement 2*). After intravenous administration of rhDAO-R568S/R571T, rhDAO-R568S/K575T, and the Fc fusion protein Fc-R568S/R571T, the corresponding DAO activity could be readily measured for 28 hr (*Figure 5*). Diamine oxidase concentrations and activities are highly correlated with $R^2$ values of >96% (*Figure 5—figure supplement 1*).

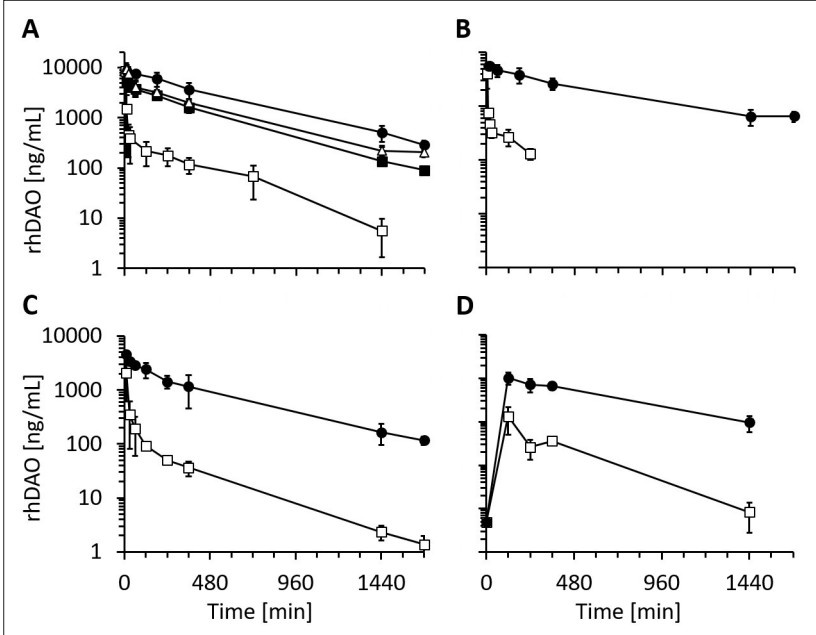

**Figure 4.** Recombinant human diamine oxidase (rhDAO) heparin-binding motif (HBM) mutants show strongly reduced clearance compared to the wildtype protein in rats and mice. (**A, B**) Reduced clearance of HBM mutants in rats. Means of the measured values ± standard deviation (SD) after administration of 1 mg/kg purified rhDAO: (**A**) WT (n = 9; □), -R568S (n = 4; ■), -R568S/R571T (n = 5; ●), -R568S/K575T (n = 4; △); (**B**) rhFcDAO (n = 6; □), Fc-R568S/R571T (n = 4; ●). (**C, D**) rhDAO-R568S/R571T mutant increases the area under the curve (AUC) more than 15-fold compared to wildtype rhDAO after intravenous and intraperitoneal injection into mice. (**C**) rhDAO-WT (□) and rhDAO-R568S/R571T (●) were injected at 1 mg/kg into the tail vein of C57BL6 mice with a body weight of about 20 g. The mean (n = 3–4 mice per time point) ± SD are shown. (**D**) rhDAO-WT and rhDAO-R568S/R571T mutant were injected intraperitoneally at 1 mg/kg in mice with a body weight of 21 g. Each time point represents the mean of 3 mice, and therefore in total 15 mice with WT and 15 mice with rhDAO-R568S/R571T were used. The means ± SD are shown. n: number of animals.

## In silico analysis of heparin binding indicates a key role of the conserved R568 residue

Comparison of the HBM in 87 DAO *Mammalia* sequences showed an overall sequence identity of 68% at positions R568 to K575, with 89, 49, and 67% conservation at positions R568, R571, and K575, respectively. In the alignment of 15 HBM sequences from old world monkeys, great apes, and humans, residues from R568 to P574 are 100% conserved. At the single variable position 575, 3 of the 15 sequences have a lysine (20%), but it is conservatively substituted by an arginine in 11 sequences (73%). The evolutionary conservation analysis is summarized in *Supplementary file 1*.

The symmetric HBM on top of the DAO dimer is composed of the residues $^{568}$RFKRKLPK$^{575}$ from both chains (*Figure 6A* and *Figure 6—figure supplement 1A*), with 10 out of 16 residues (63%) positively charged. The experimental results demonstrated that R568 is critical for heparin binding. The replacement of R568 by a serine (*Figure 6B*) was predicted to strongly decrease the binding affinity, while the variants R571T (*Figure 6C*) and K575T (*Figure 6—figure supplement 1B and C*) were predicted to be associated with lower binding energy changes (*Table 3*), supporting the experimental binding data using heparin-sepharose (*Table 1*). The squared correlation coefficient from regression analysis using affinity predictions and measured salt elution concentrations was 93% (p-value 0.0088) including wildtype and four HBM mutants but excluding the triple mutant (*Figure 6—figure supplement 2*).

In both rhDAO chains, R568 forms ionic interactions and hydrogen bonds with the sulfate groups of the heparin hexasaccharide (*Figure 6D* and *Figure 6—figure supplement 1D*). These interactions are absent in the R568S mutant (*Figure 6E* and *Figure 6—figure supplement 1E*). The R571 residue has a role in the overall architecture of the HBM since its guanidinium group forms intra-chain hydrogen bonds (*Figure 6—figure supplement 1D*), which are lost in the R571T mutant (*Figure 6—figure*

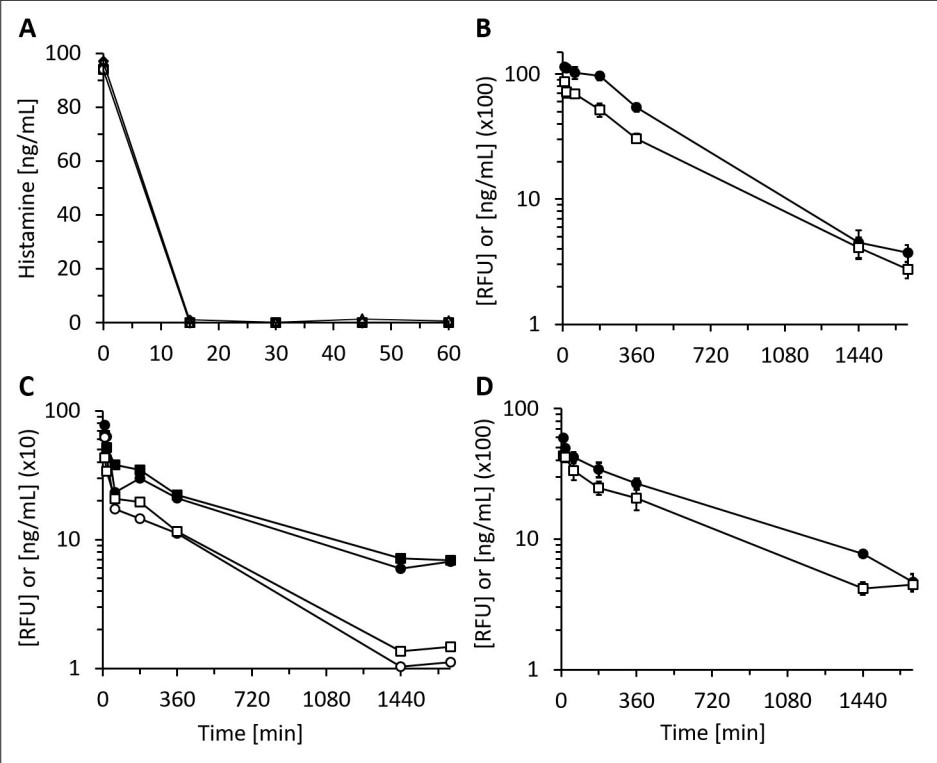

**Figure 5.** Recombinant human diamine oxidase (rhDAO) heparin-binding motif (HBM) mutants are enzymatically fully active in vitro and in vivo. (**A**) Purified rhDAO-R568S/R571T mutant rapidly and completely degrades histamine. 6 nM rhDAO-R568S/R571T with and without 100× high molecular weight heparin (HMWH) (600 nM) were incubated with 100 ng/mL histamine in 1% human serum albumin-phosphate buffered saline (HSA-PBS) and EDTA plasma of a healthy volunteer for 60 min at 37 °C. Histamine concentrations were determined in duplicate. Plasma: △; plasma with 100× HMWH: ◇; HSA-PBS: O; HSA-PBS with 100× HMWH: □. (**B–D**) rhDAO HBM mutants are enzymatically active in rats. Means of the measured values ± standard error of the mean (SEM) using 1 mg/kg rhDAO-R568S/R571T (**B**, n = 3; $R^2$ = 94.8 % with p-value<0.001, linear regression) and rhFcDAO-R568S/R571T (**D**, n = 3; $R^2$ = 98 % with p-value<0.001, linear regression) and two individually plotted measurements of rhDAO-R568S/K575T (**C**); rhDAO concentration (ng/mL; □ and O) and activity (relative fluorescence units [RFU]; ■ and ●); RFU data downscaled by a factor 50 in panels (**B**) and (**C**) and by 100 in (**D**) for easier presentation. n: number of animals.

The online version of this article includes the following figure supplement(s) for figure 5:

**Figure supplement 1.** Recombinant human diamine oxidase (rhDAO) antigen concentrations and activity are highly correlated in rat plasma.

**Figure supplement 2.** The enzymatic activity of rhDAO-R568S/R571T is comparable to rhDAO-WT.

---

supplement 1E). The negligible effect of K575T change on binding affinity can be attributed to its deep location and long distance from the heparin hexasaccharide. The destabilizing effect (*Table 3*) likely results from the loss of favorable intramolecular interactions formed by K575 (*Figure 6—figure supplement 1B and C*). Although the triple mutant K570G/R571Q/K572T (*Figure 6—figure supplement 1F and G*) showed the lowest predicted affinity of all mutants, binding to heparin-sepharose was stronger compared to R568S/R571T (*Table 1*). This discrepancy might be explained by the additive nature of the energy change calculating algorithm, which is functionally optimized for single and double point mutations and likely overestimates the effects of triple mutations (*Spassov and Yan, 2013*). Due to the surface position of R568, K570, R571, and K572, the effects of their mutations on the structural stability, even in the case of the triple mutant, are predicted to be lower (*Table 3*, *Figure 6—figure supplement 1F and G*).

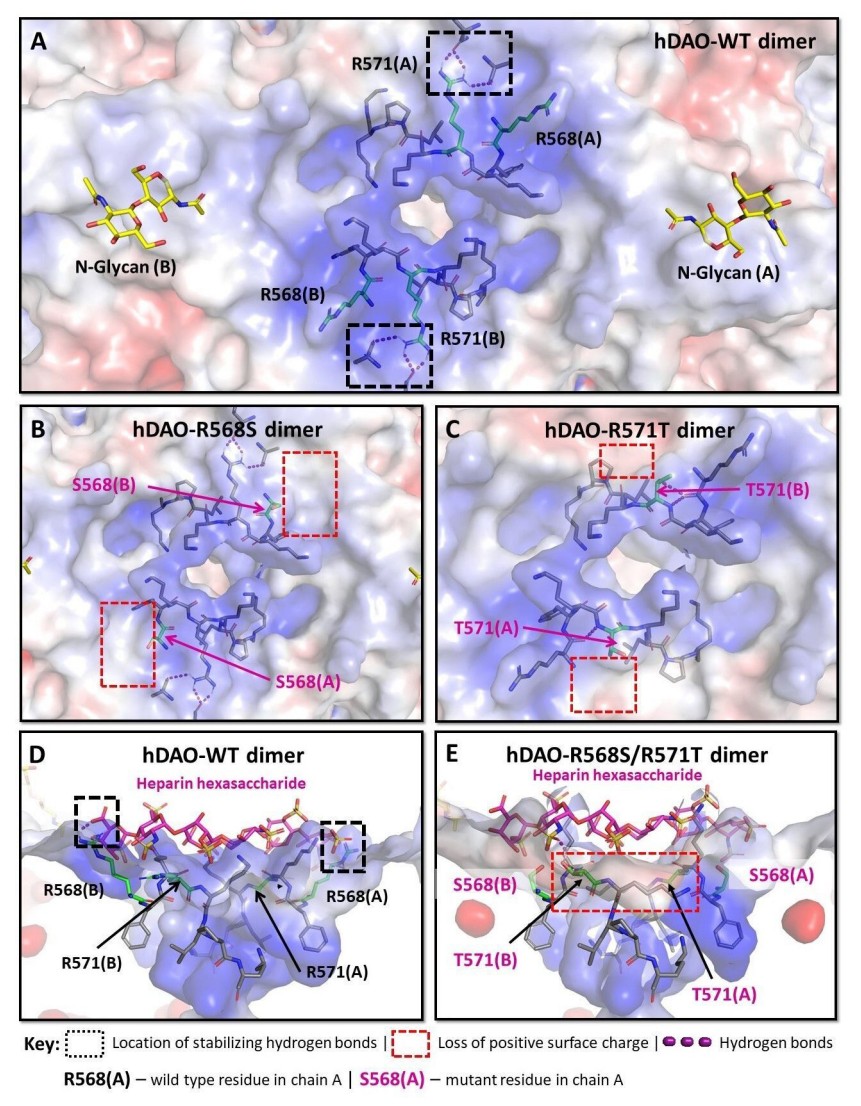

**Figure 6.** Structural analysis of the heparin-binding motif (HBM) in the 3D models for the R568S, R571T, and R568S/R571T mutants compared with the crystal structure of hDAO-WT. (**A**) Top view of the HBM in hDAO-WT. R568 and R571 (green sticks) are the key residues in the positively charged HBM formed by arginines and lysines from both chains in the hDAO-WT dimer. Intramolecular interactions formed by R571 stabilize the HBM. (**B**) Top view of the HBM in hDAO-R568S dimer. The positively charged area is reduced around S568s in the R568S mutant compared to hDAO-WT (**A**). (**C**) Top view of the HBM in hDAO-R571T. The stabilizing interactions formed by R571 in hDAO-WT (**A**) are lost, and positive patches around T571s are reduced in hDAO-R571T. (**D**) Complex of hDAO-WT dimer with heparin hexasaccharide, sliced side view. R568s form hydrogen bonds with heparin hexasaccharide. (**E**) Complex of hDAO-R568S/R571T dimer with heparin hexasaccharide, sliced side view. In the R568S/R571T mutant, the positively charged surface patches and the stabilizing interactions are lost. Red color corresponds to the negatively charged surface, blue color indicates positively charged regions. hDAO: human diamine oxidase.

The online version of this article includes the following figure supplement(s) for figure 6:

**Figure supplement 1.** Structural analysis of the heparin-binding motif (HBM) in 3D models of the K575T, K570G/R571Q/K572T, and R568S/R571T mutants versus the crystal structure of hDAO-WT.

**Figure supplement 2.** Regression analysis of in silico affinity change estimations and measured heparin-sepharose salt elution concentrations of various heparin-binding motif (HBM) mutants.

**Table 3.** Predicted free energy changes in the human diamine oxidase (hDAO) dimer by mutation of the heparin-binding motif (HBM).

| rhDAO mutant variant | Stability ΔΔG (kcal/mol) | Affinity ΔΔG (kcal/mol) |
|---|---|---|
| R568S | +0.29 | +3.50 |
| R571T | +0.73 | +1.96 |
| K575T | +2.13 | +0.55 |
| R568S/R571T | +0.60 | +5.21 |
| R568S/K575T | +2.42 | +3.93 |
| R571T/K575T | +2.86 | +2.52 |
| K570G/ R571Q/ K572T* | +1.53 | +7.77 |

The plus sign (+) in the 'Stability' column indicates the destabilizing effect on the 3D protein structure, while the plus sign (+) in the 'Affinity' column means reduced affinity to heparin hexasaccharide compared to wildtype hDAO. A higher value indicates higher effect on stability or affinity.

*Corresponds to the HBM in guinea pig, dog, rat, mouse, and Chinese hamster.

## Discussion

Plasma histamine concentrations of at least 500 ng/mL (4.5 µM) were measured in a systemic mastocytosis patient following gastrointestinal endoscopy (*Desborough et al., 1990*). We recently published data on the circulatory collapse in a mastocytosis patients with 70 ng/mL plasma histamine (*Boehm et al., 2019*). Histamine concentrations in another systemic mastocytosis patient increased from 10 ng/mL to 35 ng/mL over a few hours. The patient developed severe clinical symptoms despite high-dose treatment with the histamine receptor 1 antagonist diphenhydramine and the histamine receptor 2 antagonist ranitidine (*Boehm et al., 2019*). Despite treatment with histamine receptor antagonists up to four times the approved dose and the use of additional medications, approximately 70% of mastocytosis patients define themselves as disabled in accordance with standard definitions of disability (*Hermine et al., 2008*). Finally, 25–33% of chronic urticaria patients are resistant to histamine receptor antagonists (*van den Elzen et al., 2017*). Why is the efficacy of histamine receptor blockers during severe anaphylaxis, MC activation, or chronic urticaria limited?

*Kaliner et al., 1982* and *Owen et al., 1982* described that in healthy volunteers two combinations of histamine receptor antagonists, cimetidine with hydroxyzine and cimetidine with chlorpheniramine, respectively, were only able to sufficiently block symptom development until systemic histamine concentrations of approximately 6 ng/mL were reached. Single treatments with either histamine receptor 1 or 2 antagonist were minimally protective. The 6 ng/mL threshold is only three to four times higher compared to no treatment and 20-fold below the mean histamine concentration of 140 ng/mL reached during severe anaphylaxis after an insect sting challenge (*van der Linden et al., 1992*). Clinically relevant hypotension starts to develop at levels above 5 ng/mL histamine. We can only conclude that histamine receptor antagonists are easily overwhelmed during anaphylaxis, MC activation, and even chronic urticaria.

Herein we show for the first time that rhDAO can rapidly and completely degrade pathophysiologically relevant histamine concentrations of 100 ng/mL. Nevertheless, in rodents wildtype rhDAO showed a very fast α-distribution half-life, rendering it unsuitable for further development as a new and first-in-class biopharmaceutical. The fast clearance in vivo was independent of the asialoglycoprotein- and mannose-receptors, the two well-characterized protein clearance systems. Cellular internalization in vitro into various cell lines was dependent on the interaction with HS glycosaminoglycans and was blocked by excess heparin (*Gludovacz et al., 2020*).

We therefore mutated the putative HBM (*Novotny et al., 1994*) and unequivocally demonstrated that this 2 × 8 linear amino acid stretch, which forms a distinct positively charged ring structure, mediates not only cellular internalization but also rapid clearance in rodents. Mutations in the HBM eliminated heparin binding. Importantly, DAO activity of wildtype and various mutants was indistinguishable. The wildtype DAO dimer sequence excluding the HBM contains 84 arginine and 42 lysines residues, many of which are surface-exposed, but none of these positively charged residues seems critically involved in heparin binding. BLAST analysis of the RFKRKLPK motif revealed a high sequence identity of R568 and K575 in all mammalian DAO sequences analyzed. Our experimental data suggest the essential role of the conserved R568 residue in the heparin binding of DAO, which is supported by the in silico modeling comparing the interactions of wildtype and mutant DAO with the docked heparin hexasaccharide. While the single mutation of this amino acid was sufficient to achieve a comparable decrease in binding to heparin-sepharose versus the double mutants, a strong

reduction of cellular uptake was only accomplished with the double mutants. Neither ITC nor BLI detected any interaction of rhDAO-R568S/R571T with HMWH and HS. In silico modeling supports not only the biochemical and cellular in vitro but also the rats and mice in vivo data. The HBM mutant with the best in vivo pharmacokinetic profile, rhDAO-R568S/R571T, showed low increases in the Gibbs free energy concerning stability but demonstrated a strong decrease in affinity. The correlation coefficient comparing in silico-estimated affinity data changes with experimental heparin-sepharose in vitro binding data of wildtype and four HBM mutants is 0.93 with a p-value of less than 0.01.

Incubation of various cell lines with rhDAO-R568S/R571T did not completely abrogate cellular uptake, but flow cytometry and western blot analyses showed that at least 15% of the internalization cannot be attributed to HSPG interaction. This is in accordance with our previous study, where a 100-fold excess of HMWH over rhDAO-WT could not reduce the uptake below 5 and 25% in various cell lines (*Gludovacz et al., 2020*). In the same study, we observed the presence of high-affinity-binding sites in HUVEC/TERT2 cells in addition to the low-affinity HSPG-binding sites. Similarly, cellular binding of amyloid protein precursor and fibroblast growth factor 2 is predominantly mediated by low-affinity HSPG interactions that serve as scaffolds or co-receptors that promote and/or stabilize the formation of complexes of proteins and high-affinity receptors (*Duchesne et al., 2012*; *Reinhard et al., 2013*; *Thompson et al., 1994*; *Xu and Esko, 2014*). Since the HS-binding sites can outnumber the amount of protein-specific receptors by 100- to 1000-fold (*Duchesne et al., 2012*), it is not surprising that only a low proportion of binding can be attributed to the latter. These high-affinity-binding sites might be responsible for the β-elimination of the heparin-binding mutants. Our data are in agreement with this hypothesis. While the clearance of more than 90% of the rhDAO-WT dose in the fast α-elimination phase of less than 5 min is fully abrogated by the double mutation of R568 and R571, the long β-elimination phase of about 6 hr is more or less unchanged. It is therefore likely that this second phase is determined by a different clearance mechanism independent of the heparin-binding domain.

It will be interesting to study the distribution of wildtype versus HBM mutant DAO variants after intravenous infusion in animal models because it is not clear which cells or organs remove 90% of the wildtype DAO protein within a few minutes. The liver has been considered mainly responsible for elimination of DAO, but in view of the dependence of DAO internalization on HS, endothelial cells throughout the circulatory system could rapidly remove DAO (*D'Agostino et al., 1986*). These cells abundantly express HSPGs and are able to bind and internalize DAO in vitro (*Fuster and Wang, 2010*; *Gludovacz et al., 2020*).

The half-life of the HBM mutant rhDAO-R568S/R571T is approximately 6 hr in rodents, which is certainly sufficient for the treatment of most human anaphylaxis events. The rodent HBM does not contain the central $^{570}$KRK$^{572}$ sequence present in humans but instead $^{570}$GQT$^{572}$ and therefore the half-life in humans could be longer, assuming that the 6 hr half-life is still at least partially determined by the same HS internalization process. The rodent HBM is only weakly heparin binding.

In conclusion, mutations in the proposed and now proven HBM converted rhDAO wildtype protein into a candidate for a first-in-class histamine receptor-independent biopharmaceutical for the rapid and complete elimination of excessive histamine. First clinical indications might be diseases where it has been known for decades that histamine plays an important pathophysiological role. These include anaphylaxis, MC activation syndrome, mastocytosis, and chronic urticaria. Nevertheless, the use of HBM mutants will also allow clinical proof-of-concept studies in other diseases, where the involvement of histamine and MCs is suspected, but where clinical studies with available histamine receptor antagonists were unsuccessful. No drug blocking histamine receptor 4 has been approved to date. This group of diseases includes, amongst others, asthma, atopic dermatitis, infusion reactions, different forms of pruritus, and inflammatory bowel disease. rhDAO with HBM mutations might overcome the current limitations of histamine receptor antagonists for the treatment of diseases that lead to life-threatening histamine exposure but are resistant to conventional treatment modalities.

## Acknowledgements

The authors thank Evercyte GmbH for providing various cell lines for the study. Irene Schaffner and Jakob Wallner of the Core Facility 'Biomolecular & Cellular Analysis' (University of Natural Resources and Life Sciences) are greatly acknowledged for their support with automated ITC and BLI analyses and the Core Facility 'Multiscale Imaging' (University of Natural Resources and Life Sciences) for technical support with microscopy. We thank Leander Sützl for the selection and alignment of mammalian DAO

sequences. We are grateful to the bioinformatics (J.V. Lehtonen), translational activities and structural biology (FINStruct) infrastructure support from Biocenter Finland and CSC IT Center for Science for computational infrastructure support at the Structural Bioinformatics Laboratory (SBL), Åbo Akademi University. SBL is part of the NordForsk Nordic POP (Patient Oriented Products), the Solutions for Health strategic area of Åbo Akademi University, and the InFlames Flagship program of the Academy of Finland on inflammation, cancer and infection, University of Turku and Åbo Akademi University. We are indebted to Sarah Ely for the final polish in the proper usage of the English language. The publication fees were covered by the BOKU Vienna Open Access Publishing Fund.

## Additional information

### Competing interests

Elisabeth Gludovacz, Bernd Jilma, Nicole Borth, Thomas Boehm: is named as an inventor with The Medical University of Vienna and the University of Natural Resources and Life Sciences of a patent describing the rhDAO heparin-binding motif mutants presented herein (patent pending WO2020169577A1). The other authors declare that no competing interests exist.

### Funding

| Funder | Grant reference number | Author |
|---|---|---|
| Austrian Science Fund | T1135 | Elisabeth Gludovacz |
| Sigrid Juséliuksen Säätiö | | Serhii Vakal<br>Tiina A Salminen |
| Medicinska Understödsföreningen Liv och Hälsa | | Serhii Vakal<br>Tiina A Salminen |

The funders had no role in study design, data collection and interpretation, or the decision to submit the work for publication.

### Author contributions

Elisabeth Gludovacz, Conceptualization, Data curation, Formal analysis, Funding acquisition, Investigation, Methodology, Project administration, Supervision, Validation, Visualization, Writing – original draft, Writing – review and editing; Kornelia Schuetzenberger, Marlene Resch, Katharina Tillmann, Sigrid Vondra, Investigation, Methodology; Karin Petroczi, Markus Schosserer, Investigation; Serhii Vakal, Tiina A Salminen, Investigation, Methodology, Visualization, Writing – review and editing; Gerald Klanert, Investigation, Validation; Jürgen Pollheimer, Writing – review and editing; Bernd Jilma, Conceptualization, Funding acquisition, Supervision, Writing – review and editing; Nicole Borth, Funding acquisition, Supervision, Writing – review and editing; Thomas Boehm, Conceptualization, Data curation, Formal analysis, Funding acquisition, Methodology, Project administration, Supervision, Validation, Visualization, Writing – original draft, Writing – review and editing

### Author ORCIDs

Elisabeth Gludovacz http://orcid.org/0000-0002-1837-2422
Markus Schosserer http://orcid.org/0000-0003-2025-0739
Jürgen Pollheimer http://orcid.org/0000-0001-8440-5221
Thomas Boehm http://orcid.org/0000-0002-8294-0797

### Ethics

The experimental protocols for the treatment of rats and mice were approved by the local Animal Welfare Committee and the Federal Ministry of Science, Research and Economy (GZ 66.009/0152--WF/V/3b/2014) and conducted in full accordance with the ARRIVE guidelines.

### Decision letter and Author response

Decision letter https://doi.org/10.7554/eLife.68542.sa1
Author response https://doi.org/10.7554/eLife.68542.sa2

# Additional files

## Supplementary files
• Supplementary file 1. The evolutionary conservation analysis of the diamine oxidase (DAO) heparin-binding motif is summarized in *Supplementary file 1*.

• Transparent reporting form

## Data availability
All data generated or analysed during this study are included in the manuscript and supporting files. Source data files have been provided for Figures 2 and 3.

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
