## [Decision Letter]

**Acceptance summary:**

The human enzyme diamine oxidase (hDAO) can rapidly degrade histamine, but unfortunately is cleared from blood too rapidly after injection. The authors have altered key amino acid residues in hDAO responsible for this rapid clearance, switching it into a form which has long lasting effects on histamine clearance in plasma. This recombinant protein could now be tested in clinical trials as a biopharmaceutical.

**Decision letter after peer review:**

Thank you for submitting your article "Heparin-binding motif mutations of human diamine oxidase allow the development of a first-in-class histamine-degrading biopharmaceutical" for consideration by *eLife*. Your article has been reviewed by 2 peer reviewers, and the evaluation has been overseen by a Reviewing Editor and Betty Diamond as the Senior Editor. The following individuals involved in review of your submission have agreed to reveal their identity: Jeremy Turnbull (Reviewer #1); Ding Xu (Reviewer #2).

Essential revisions:

1. The study lacks a key in vivo experiment to demonstrate the translational potential of the HBM mutant. It is true that the clearance of the mutant was greatly reduced, but the authors need to show that the mutant really works better than WT in vivo. This experiment should not be hard to carry out, and can be as simple as injecting animals with histamine, and then inject WT versus mutant DAO followed by measuring remaining histamine concentration in the plasma at different time points.

2. It is uncommon to mutate basic residues to serine and threonine because these polar residues can still potentially form hydrogen bonds with HS. Have the authors checked alanine mutations (most commonly used to screen HS-binding residues) or glutamine mutations (which electrostatically repels HS) of these residues. There is a good chance that these mutants might give even better reduction in HS-binding and improved clearance profile.

3. To ensure that the enzymatic activity of the HBM mutant was not affected, the authors need to perform an activity assay with dose range of DAO. The current dose (Figure 5A) at 6 nM is apparently too high to detect any potential difference in activity between WT and mutants because all substrate was fully degraded within 15 mins, and there was no time points before 15 mins.

4. Figure 5 and text – some comparison in vivo data with WT protein would be a valuable addition.

5. Figure 1B: more explanation of why some error bars are missing is needed for clarity (not just that the amounts were >500nM).

6. Table 1: should state peak concentration for elution.

7. Table 2: its not clear why for the mice data it says "Fold-increase WT etc.." in the middle of the section in table.

8. Figure 4: difference between A and B not clear from legend.

9. Most panels (B-G) in Figure 6 are unnecessary and confusing. I can't really see what is the point of panel B and C and what all these labeled distances for. Panel D and E were shown in two different orientation and scales, which really didn't help understanding. Panel F and G are totally unnecessary.

10. Table I, it's not very meaningful, and can be misleading, to express the mutant binding to heparin Sepharose as a % binding of WT. The proper way is to list the reduction in NaCl concentration that is required for elusion.

11. The Discussion can be much more concise and the authors should limit the discussion to points directly related to the current study. For instance, the discussion about potential anti-drug antibodies was not necessary.*Reviewer #1:*

High histamine concentrations in plasma are an issue in certain clinical settings, and anti-histamines cannot always rescue this. The human enzyme diamine oxidase (hDAO) can rapidly degrade histamine, but unfortunately is cleared from blood too rapidly after injection. In this study the authors aimed to alter key amino acid residues in hDAO responsible for this rapid clearance, with the goal of creating a form which has long lasting effects on histamine clearance in plasma. The manuscripts strengths are a coherent and convincing combination of biochemical, cellular, molecular modelling and in vivo pharmacokinetic and activity data. Weaknesses are a few minor issues with clarity of some figure legends and tables, and possible need for additional in vivo data on wild type protein. Overall the authors claims are well supported and indicate that this recombinant protein could now be tested in clinical trials as a biopharmaceutical.

*Reviewer #2:*

In this manuscript the authors aimed to find a way to improve the pharmacokinetics of human diamine oxidase (DAO), an enzyme that rapidly degrades histamine. Although DAO has a clear potential for treating diseases with excessive histamine, the clearance rate of recombinant hDAO was extremely fast (5 min), which severely hinders the effectiveness of the treatment. Previously the group has identified cell surface heparan sulfate (HS) as the main uptake receptor for hDAO, which might also play a role in the rapid clearance of hDAO in vivo. They have shown convincingly that by mutating several putative HS-binding residues of hDAO, the clearance rate of hDAO was dramatically improved to 6 hours. These mutations appeared to have no negative impact on hDAO activity, however this point will need to be more rigorously tested. This study will be a nice example showing that manipulating protein-HS interactions can have profound impact on the pharmacokinetics of therapeutically useful HS-binding proteins. In the current form however, the study lacks a key in vivo experiment to demonstrate whether the mutant truly works better than WT hDAO.

---

## [Author Response]

Essential revisions:1. The study lacks a key in vivo experiment to demonstrate the translational potential of the HBM mutant. It is true that the clearance of the mutant was greatly reduced, but the authors need to show that the mutant really works better than WT in vivo. This experiment should not be hard to carry out, and can be as simple as injecting animals with histamine, and then inject WT versus mutant DAO followed by measuring remaining histamine concentration in the plasma at different time points.

We tend not to entirely agree with this statement for several reasons. This is not a competition between wildtype DAO and heparin-binding motif mutated DAO. Wildtype DAO is not suitable for further development considering that GMP manufacturing costs several million $/€. It is possible to only move one variant to the next development stage and it will certainly not be the wildtype DAO variant. Enzymes show in most cases a clear dose response (we have shown this with DAO many times in suitable substrate and enzyme concentration ranges) and as you can see from the submitted data wildtype DAO concentrations are manifold lower within a few minutes compared to mutated variants and therefore wildtype DAO can only degrade a fraction of the histamine compared to mutated DAO. It is somewhat unethical to perform an animal experiment, where the result is clear before the experiment is done and this is also the reason, why we never did these kinds of experiments. The knowledge gain is minimal and as mentioned above patients will never be treated with wildtype DAO. Our animal experiment protocol expired a few months ago and writing a new protocol and receiving approval would take at least 6 to 9 months. This effort with minimal scientific gain is not an optimal use of anyway constrained resources. The ethics committee might also not allow such experiments, because if there is no or a minimal concentration of enzyme, histamine cannot be degraded and testing/killing animals with no clear scientific gain/rational is unethical.

Nevertheless, we totally agree that the mutated enzyme should be tested for its ability to rapidly degrade histamine in vivo or modify histamine-dependent symptoms in animal models, but not necessarily in comparison with wildtype DAO. We are currently writing an animal protocol doing such experiments. Unfortunately, clear histamine dependent pathologies in mice/rats are not really available because it seems that other mediators like PAF and serotonin play a more prominent role, e.g. in anaphylaxis. There are no animal models in chronic urticaria or mast cell activation syndrome, two other human diseases, where histamine plays an important role. It might sound easy to inject histamine into mice but the kidney removes 50% of circulating histamine in two minutes under normal kidney function and histamine sensitivity of rodents is at least 100- to 1000-fold lower and therefore high concentrations of histamine are necessary to induce symptoms. We know from many experiments using human plasma that once we have reasonable DAO concentrations present we cannot measure histamine anymore, because it is rapidly degraded.

We guess it is an editorial decision because for the reasons expressed above we have difficulties performing the proposed experiment. The other reviewer seems to concur using “possible need for additional in vivo data”.

2. It is uncommon to mutate basic residues to serine and threonine because these polar residues can still potentially form hydrogen bonds with HS. Have the authors checked alanine mutations (most commonly used to screen HS-binding residues) or glutamine mutations (which electrostatically repels HS) of these residues. There is a good chance that these mutants might give even better reduction in HS-binding and improved clearance profile.

One to two positively charged amino acids of the putative heparin-binding motif (F)RFKRKLPK at the amino acids 568-575 were replaced by slightly polar serine or threonine that potentially preserve the structural integrity of the protein. Threonine and serine have been used to replace arginine residues in the heparin-binding site of fibronectin, resulting in significantly reduced binding affinities, while at the same time retaining the protein’s three-dimensional structure (Busby et al., 1995; Kapila et al., 2001). The replacement with these slightly polar amino acids resulted in two mutants (R571T and R571T/K575T) that could not be successfully expressed in CHO cells. Hence, the exchange of the positively charged amino acids with non-polar alanine could potentially disrupt the local protein fold. Glutamine might behave similarly and might be also more immunogenic compared to threonine or serine due to its larger size.

We agree that the exchange with alanine would be scientifically interesting, but we believe that our study shows that the exchange of positively charged amino acids in the heparin-binding motif of DAO with serine and threonine more or less completely inhibits the interaction with heparin and heparan sulfate and cellular uptake. The α‑distribution half-life is eliminated using the arginine double mutant and therefore it is not clear that clearance could be further reduced using other mutants. The β‑elimination half-life is similar to wildtype DAO arguing against the involvement of the heparin binding motif in the β‑elimination half-life and therefore it is unlikely that clearance might be changed.

We might not have pointed this out sufficiently in the Discussion section and have therefore added the following text (lines 267-275 and 282-300):

“BLAST analysis of the RFKRKLPK motif revealed a high sequence identity of R568 and K575 in all mammalian DAO sequences analyzed. Our experimental data suggest the essential role of the conserved R568 residue in the heparin binding of DAO, which is supported by the *in silico* modeling comparing the interactions of wildtype and mutant DAO with the docked heparin hexasaccharide. While the single mutation of this amino acid was sufficient to achieve a comparable decrease in binding to heparin-sepharose versus the double mutants, a strong reduction of cellular uptake was only accomplished with the double mutants. Neither ITC, nor BLI detected any interaction of rhDAO-R568S/R571T with HMWH and HS…………… Incubation of various cell lines with rhDAO-R568S/R571T did not completely abrogate cellular uptake but flow cytometry and western blot analyses showed that at least 15% of the internalization cannot be attributed to HSPG interaction. This is in accordance with our previous study, where a 100-fold excess of HMWH over rhDAO-WT could not reduce the uptake below 5% and 25% in various cell lines (Gludovacz et al., 2020). In the same study we observed the presence of high affinity binding sites in HUVEC/TERT2 cells in addition to the low affinity HSPG binding sites. Similarly, cellular binding of Amyloid Protein Precursor and Fibroblast Growth Factor 2 is predominantly mediated by low affinity HSPG interactions that serve as scaffolds or co-receptors that promote and/or stabilize the formation of complexes of proteins and high-affinity receptors (Duchesne et al., 2012; Reinhard et al., 2013; Thompson et al., 1994; Xu and Esko, 2014). Since the HS-binding sites can outnumber the amount of protein-specific receptors by 100- to 1000-fold (Duchesne et al., 2012), it is not surprising that only a low proportion of binding can be attributed to the latter. These high affinity binding sites might be responsible for the β‑elimination of the heparin-binding mutants. Our data are in agreement with this hypothesis. While the clearance of more than 90% of the rhDAO-WT dose in the fast α‑elimination phase of less than five minutes is fully abrogated by the double mutation of R568 and R571, the long β‑elimination phase of about six hours is more or less unchanged. It is therefore likely that this second phase is determined by a different clearance mechanism independent of the heparin binding domain.”

Busby TF, Argraves WS, Brew SA, Pechik I, Gilliland GL, Ingham KC. 1995. Heparin Binding by Fibronectin Module III-13 Involves Six Discontinuous Basic Residues Brought Together to Form a Cationic Cradle (∗). *Journal of Biological Chemistry* 270:18558–18562. doi:10.1074/jbc.270.31.18558

Duchesne L, Octeau V, Bearon RN, Beckett A, Prior IA, Lounis B, Fernig DG. 2012. Transport of Fibroblast Growth Factor 2 in the Pericellular Matrix Is Controlled by the Spatial Distribution of Its Binding Sites in Heparan Sulfate. *PLOS Biology* 10:e1001361. doi:10.1371/journal.pbio.1001361

Gludovacz E, Schuetzenberger K, Resch M, Tillmann K, Petroczi K, Vondra S, Vakal S, Schosserer M, Virgolini N, Pollheimer J, Salminen TA, Jilma B, Borth N, Boehm T. 2020. Human diamine oxidase cellular binding and internalization in vitro and rapid clearance in vivo are not mediated by N-glycans but by heparan sulfate proteoglycan interactions. *Glycobiology*. doi:10.1093/glycob/cwaa090

Kapila Y, Doan D, Tafolla E, Fletterick R. 2001. Three-dimensional structural analysis of fibronectin heparin-binding domain mutations. *J Cell Biochem Suppl* Suppl 36:156–161. doi:10.1002/jcb.1095

Reinhard C, Borgers M, David G, Strooper BD. 2013. Soluble amyloid-β precursor protein binds its cell surface receptor in a cooperative fashion with glypican and syndecan proteoglycans. *J Cell Sci* 126:4856–4861. doi:10.1242/jcs.137919

Thompson LD, Pantoliano MW, Springer BA. 1994. Energetic Characterization of the Basic Fibroblast Growth Factor-Heparin Interaction: Identification of the Heparin Binding Domain. *Biochemistry* 33:3831–3840. doi:10.1021/bi00179a006

Xu D, Esko JD. 2014. Demystifying Heparan Sulfate–Protein Interactions. *Annual Review of Biochemistry* 83:129–157. doi:10.1146/annurev-biochem-060713-035314

3. To ensure that the enzymatic activity of the HBM mutant was not affected, the authors need to perform an activity assay with dose range of DAO. The current dose (Figure 5A) at 6 nM is apparently too high to detect any potential difference in activity between WT and mutants because all substrate was fully degraded within 15 mins, and there was no time points before 15 mins.

Again, it is not a competition between wildtype and mutant DAO because the wildtype DAO variant will never be further developed for human use. Even if the enzymatic activity of the heparin binding motif mutants were lower (which is unlikely the case = see below), one would simple use more enzyme. The necessary dosage of rhDAO enzyme administered to human patients can only be determined in appropriate Phase 2 studies, where the primary endpoint is not histamine degradation (a secondary endpoint), but symptom reduction. Therefore, it is not clear whether 6 nM or 1 µg/ml DAO is too low or too high. For rapid elimination it might be too low but for chronic treatment too high!

Nevertheless, we measured DAO activity many times, comparing wildtype with heparin-binding mutants and added one example to the supplement. Even at almost 10-fold lower DAO concentrations (we tested 6, 2 and 0.7 nM or 1, 0.33 or 0.11 µg/mL) we did not see any relevant differences in the enzymatic activity between wildtype and mutated DAO variants (see Figure 5 —figure supplement 2). Based on the published 3D structure it is also not really surprising that we did not see any activity differences, because the active center and the substrate channel are “far away” from the heparin binding motif. Nevertheless, we might have missed minor effects.

4. Figure 5 and text – some comparison in vivo data with WT protein would be a valuable addition.

See comments to point 1.

5. Figure 1B: more explanation of why some error bars are missing is needed for clarity (not just that the amounts were >500nM).

The measured values where outside the standard curve and we did not remeasure the samples because the interpretation of the results would not change. We do not really understand, why the clarity would change, if the values were 550 nM or 600 nM and if error bars were shown. These values above 500 nM are compared to values <5 nM, at least a 100-fold difference. The results would be under any circumstances statistically highly significant, which we did not calculate/show, because it is “obvious”. One could derive error bars from the absorption values.

6. Table 1: should state peak concentration for elution.

The requested information was added to the table legend.

7. Table 2: its not clear why for the mice data it says "Fold-increase WT etc.." in the middle of the section in table.

We revised the table.

8. Figure 4: difference between A and B not clear from legend.

We made the necessary corrections.

9. Most panels (B-G) in Figure 6 are unnecessary and confusing. I can't really see what is the point of panel B and C and what all these labeled distances for. Panel D and E were shown in two different orientation and scales, which really didn't help understanding. Panel F and G are totally unnecessary.

We have thoroughly revised the figure. Now, panel A shows the positively charged heparin-binding motif (HBM; blue surface) in the hDAO-WT dimer and highlights the essential residues for heparin binding; panel B demonstrates the loss of the positively charged surface in the R568S mutation; panel C illustrates that the R571T mutation reduces the positive charge and eliminates stabilizing polar interactions within the HBM in the hDAO-R571T dimer; panel D shows the predicted binding mode between hDAO-WT dimer and heparin hexasaccharide; panel E shows a dramatic loss of positive charge in hDAO-R568S/R571T dimer and stabilizing interactions with the heparin hexasaccharide.

To draw the reader’s attention to the main point of each figure, we added dashed black frames to the specific polar interactions of hDAO-WT and dashed red frames to the regions where a positive surface charge is lost in the mutants. The labels for bonds were removed, and the stick width for residues was reduced to make the pictures clearer. According to the reviewer’s suggestion, we removed panels F and G from Figure 6 and incorporated them into Figure 6 —figure supplement 1. Now, Figure 6 —figure supplement 1F and 1G respectively show specific interactions of residues 568 and 571 in hDAO-WT and hDAO-R568S/R571T mutant. From the 2D diagrams (panels C, D, F, and G), one can easily see which interactions are lost due to a specific mutation.

10. Table I, it's not very meaningful, and can be misleading, to express the mutant binding to heparin Sepharose as a % binding of WT. The proper way is to list the reduction in NaCl concentration that is required for elusion.

We made the requested alterations in Table 1 and Figure 6 —figure supplement 2.

11. The Discussion can be much more concise and the authors should limit the discussion to points directly related to the current study. For instance, the discussion about potential anti-drug antibodies was not necessary.

We revised the Discussion section.